# Fibres and Colorectal Cancer: Clinical and Molecular Evidence

**DOI:** 10.3390/ijms241713501

**Published:** 2023-08-31

**Authors:** Francesca Celiberto, Adriana Aloisio, Bruna Girardi, Maria Pricci, Andrea Iannone, Francesco Russo, Giuseppe Riezzo, Benedetta D’Attoma, Enzo Ierardi, Giuseppe Losurdo, Alfredo Di Leo

**Affiliations:** 1Section of Gastroenterology, Department of Precision and Regenerative Medicine and Ionian Area, University of Bari, 70124 Bari, Italy; 2Ph.D. Course in Organs and Tissues Transplantation and Cellular Therapies, Department of Precision and Regenerative Medicine and Ionian Area, University of Bari, 70124 Bari, Italy; 3THD s.p.a., 42015 Correggio, Italy; 4Functional Gastrointestinal Disorders Research Group, National Institute of Gastroenterology IRCCS “Saverio de Bellis”, 70013 Castellana Grotte, Italy; francesco.russo@irccsdebellis.it (F.R.); giuseppe.riezzo@irccsdebellis.it (G.R.); benedetta.dattoma@irccsdebellis.it (B.D.)

**Keywords:** colorectal cancer, fibers, diet, chemoprevention

## Abstract

Colorectal cancer (CRC) is one of the leading causes of mortality for cancer in industrialized countries. The link between diet and CRC is well-known, and presumably CRC is the type of cancer which is most influenced by dietary habits. In Western countries, an inadequate dietary intake of fibers is endemic, and this could be a driving factor in the increase of CRC incidence. Indeed, several epidemiologic studies have elucidated an inverse relationship between daily fiber intake and risk of CRC. Long-term prognosis in CRC survivors is also dependent on dietary fibers. Several pathogenetic mechanisms may be hypothesized. Fibers may interfere with the metabolism of bile acids, which may promote colon carcinogenesis. Further, fibers are often contained in vegetables which, in turn, contain large amounts of antioxidant agents like resveratrol, polyphenols, or phytoestrogens. Moreover, fibers can be digested by commensal flora, thus producing compounds such as butyrate, which exerts an antiproliferative effect. Finally, fibers may modulate gut microbiota, whose composition has shown to be associated with CRC onset. In this regard, dietary interventions based on high-fiber-containing diets are ongoing to prevent CRC development, especially in patients with high potential for this type of tumor. Despite the fact that outcomes are preliminary, encouraging results have been observed.

## 1. Introduction: Epidemiology and Molecular Basis of Colorectal Cancer

Colorectal cancer (CRC) is one of the most common malignancies, being the third leading cause of cancer death worldwide, mostly in Europe and the United States, although there has been a rise in its incidence in Asia over recent decades. In 2018, the incidence of new cases was approximately 2 million, while it caused 1 million deaths worldwide [1].

Colorectal carcinoma is classified into sporadic (70%), inherited (5%), and familial (25%). There are three main genetic pathways that can lead to this situation: chromosomal instability (CIN), microsatellite instability (MSI), and CpG island methylator phenotype (CIMP). The CIN pathway is one of the main causes of CRC (85%) and is characterized by chromosomal imbalance that leads to loss of heterozygosity. The microsatellite instability pathway is caused by a hypermutable phenotype due to loss of DNA repair mechanisms. Epigenetic instability, which is responsible for the CpG island methylator phenotype, is another common feature in CRC. The main characteristic of CIMP tumors is the hypermethylation of oncogene promoters, which leads to genetic silencing and a loss of protein expression [2].

In the CIN scenario, some genetic alterations involved in hereditary forms of CRC are also responsible for sporadic forms. In particular, Wnt/β-catenin, TGF-β receptor, Notch, and Hedgehog pathways are involved.

The Wnt/β-catenin pathway is influenced by mutations of the APC gene, which generates an incorrect translocation of the protein β-catenin to the nucleus. The APC gene encodes for a specific tumor suppressor that creates the destruction complex, in association with axin, a cyclin-dependent kinase inhibitor (CKI), and glycogen synthase kinase 3 (GSK-3), a serine/threonine protein kinase [3].

This complex generates a β-TrCP (protein involved in the regulation of the cell cycle) [4] recognition site by phosphorylation of a conserved Ser/Thr-rich sequence near the β-catenin amino terminus. This process requires scaffolding of the kinases and β-catenin by axin. Ubiquitinated β-catenin is degraded by the proteasome. The molecular mechanisms underlining several aspects of destruction complex function are poorly understood, especially the role of APC [5].

The interaction between Frizzled (Fz), a transmembrane receptor, and its ligand Wnt, regulates β-catenin synthesis. Accumulation of β-catenin in the nucleus leads to an interaction with many Transcription Factors, as a way to modulate mitosis and cell proliferation. Mutations of the APC gene generate an abnormal accumulation of β-catenin in the nucleus. The consequence is an incorrect activation of transcription of Wnt target genes, which leads to tumoral cell proliferation.

TGF-β/SMAD is a signaling pathway involved in cell proliferation, differentiation, and apoptosis. Ligand TGF-β interacts with type II TGF-β receptors. Consequently, the receptor phosphorylates type SMAD Transcription Factors, SMAD2 and SMAD3. These Transcription Factors create a complex with SMAD4 and migrate in the nucleus, in order to promote cyclin and apoptosis-associated proteins. TGF-β has a tumor suppressor role. Anyway, alterations in the expression of the gene encoding for TGF-β receptor can promote tumoral cell proliferation.

Notch and Hedgehog pathways are linked to gut epithelium proliferation. The Notch pathway is based on the link between Notch ligands and their transmembrane receptors. These receptors migrate in the nucleus in order to regulate the determination of cell fate, cell differentiation, and oncogenesis. Anyway, in the case of cancer, point mutations and genetic alteration can constitutively activate this pathway, leading to indiscriminate cell proliferation.

Finally, the Hedgehog signaling pathway is associated with protein ligands Indian, Sonic, and Desert. Their receptor is the Patched protein, which suppresses the activity of Smoothened, a G protein-coupled receptor. This signaling pathway regulates the expression of many target genes. Abnormal activation of Hedgehog signaling is linked to the development of CRC [3].

## 2. Dietary Fiber

Dietary fibers are an environmental factor associated with risk for colorectal cancer. The effects of their ingestion might impact the complex events that characterize colorectal oncogenesis [5]. The definition of dietary fiber includes the non-digestible forms of polysaccharides and lignin found in vegetables. Among dietary fibers there are non-starch polysaccharides, pectins, hemicelluloses, cellulose hydro-colloids, and fructo-oligosaccharides [6]. In terms of structure, polysaccharides are categorized into linear or nonlinear molecules [7].

Moreover, functional fibers are defined as isolated, non-digestible carbohydrates (for example, inulin and oligofructose) with positive physiological effects for the human organism. Another classification of dietary fibers is based on their water solubility (Table 1). Soluble fibers can be found in vegetables and fruits, while cereals are sources of insoluble fibers. The gut microbiota acts on dietary fiber with fermentation; soluble fibers ferment more rapidly than insoluble fibers [8,9].

The correct fiber intake for adults is 30–35 g per day for men and 25–32 g per day for women, bringing benefits to the gut microbiome and metabolic health, as well as reducing cardiovascular diseases and the risk of colonic cancer [8].

Major food sources of dietary fiber are cereals/grains, vegetables, fruits, and legumes. Dietary fibers from different food sources are heterogeneous with respect to chemical composition, physicochemical properties, and solubility and thus are hypothesized to present varying degrees of anticarcinogenic properties. However, the evidence on the relationships of dietary fiber intake from different food sources with colorectal cancer and adenoma risks is mixed. Further, little is known about the shape of the relationships as well as the relative importance of fiber source [10].

The mechanisms relating diet with CRC are complex and still poorly understood. A diet with a high intake of whole grains, fruits, and vegetables is accompanied by the intake of lower-energy-density foods and lower intake of foods with high glycemic index, glycemic load, and fat, which has been shown to be related to decreased risk of obesity and diabetes, both identified risk factors for CRC. Nonetheless, studies that had adjusted the risk for obesity and diabetes reported that associations persisted in adjusted analyses. Further possible explanations for the reduced risk of CRC and intake of these food groups might be driven by the high content of fiber, also including resistant starch, oligosaccharides, and lignins, which are related to increased stool mass, decreasing colonic transit time, prebiotic effects such as action of bacterial enzymes, and fecal bile acid concentration, and are suggested to play a role in colorectal carcinogenesis. Moreover, whole grains, fruits, and vegetables are a source of minerals (e.g., magnesium), and, in particular, fruit contains a wide range of antioxidant vitamins, flavonoids, and carotenoids, and studies suggested a potential protective role against CRC [11].

Moreover, dietary fiber may influence the composition of gut microflora. The human gut contains a complex microbial community, composing the microbiota. By means of metagenomic, meta-transcriptomic, and bioinformatics tools, it is possible to define the microbial population, to monitor its compositional changes, and to understand how microbial communities interact with the gut environment. The healthy adult human gut microbiota consists of about 10^13^ microorganisms of different species, with significant abundance of Firmicutes and Bacteroidetes phyla [12]. Table 1 reports the classification of soluble and insoluble fibers.
ijms-24-13501-t001_Table 1Table 1Differences between soluble and insoluble fibers [13].Soluble FibersInsoluble FibersFound in vegetables and fruitsFound in cerealsDissolves in water and gastrointestinal fluids when it enters the stomach and intestines. It is transformed into a gel-like substance, which is digested by bacteria in the large intestine, releasing gases and a few calories.Does not dissolve in water or gastrointestinal fluids and remains more or less unchanged as it moves through the digestive tract. Because it is not digested at all, insoluble fiber is not a source of calories.Rapid fermentationSlow fermentationLow fat absorption and cholesterol levels, reducing the risk of cardiovascular diseasePrevent constipation, speeding up gut movementsStabilize blood sugar levelsReduce the risk of cardiovascular disease


## 3. Interaction between Fiber and the Microbiota

Dietary formulation can impact gut microbiota composition, diversity, and richness. In particular, dietary fiber provides various substrates for fermentation reactions carried out by species of microbes that have specific enzymes to degrade these complex carbohydrates [14]. In fact, bacteria use dietary fibers as a source of energy, metabolizing them through fermentation [12].

This biochemical process, operated by the gut flora, gives Short-Chain Fatty Acids (SCFAs) as products, including butyrate, acetate, and propionate [12,14]. These kinds of fatty acids are able to lower pH in the colon, preventing the conversion of bile acid metabolites into more toxic forms. In particular, butyrate is an anti-carcinogenic compound, able to reduce cell proliferation and induce apoptosis [10,15].

For example, three different fiber-rich diets were given to growing pigs, one replacing corn, soybean meal, and soybean oil in the diet with 20% sugar beet pulp (SBP), defatted rice bran (DFRB), or soybean hull (SBH). The analysis of fresh feces demonstrated that principal coordinate analysis (PCoA) revealed distinctly different microbial communities on the DFRB diet and SBH diet across different times; additionally, the stool microbiota of the four diet groups displayed remarkably dissimilar clusters at each time point. With adaptation time increased from 7 to 21 d, cellulose-degrading bacteria and SCFA-producing bacteria (in particular *Ruminococcaceae* _UCG-014, *Rikenellaceae* _RC9_gut_group, and *Bifidobacterium*) increased in the fiber inclusion diets, and pathogenic genera (e.g., *Streptococcus* and *Selenomonas*) were increased in the basal diet. Moreover, the intestinal flora of growing pigs adapted more easily and quickly to the SBP diet compared to the DFRB diet, as reflected by the concentration of propionate, butyrate, isovalerate, and total SCFA [16].

Hydrogen sulfide is produced by gut microbiota, and it has been proposed as a driver for CRC. A study demonstrated how the changes in gut microbiota composition induced by fibers may reduce the microbial production of hydrogen sulfide, thus reducing the risk of CRC [17]. The enrichment of gut flora of beneficial commensals (such as Clostridiales, Blautia, Roseburia), phenolic metabolites (benzoate and catechol metabolism), and dietary components (ferulic acid-4 sulfate, trigonelline, and salicylate) was correlated with anti-CRC efficacy; the rice bran dietary treatment showed a ∼72% reduction in the incidence of colonic epithelial erosion in a mouse model of inflammation-induced carcinogenesis in this regard [18]. In a CRC cell culture model consisting of human HCT-116 cells, it was demonstrated that treatment with bile acids increased proliferation by dysregulating forty-one apoptosis-related genes (including signaling pathways) with greater than onefold change. Moreover, butyrate treatment inhibited bile acid-stimulated cell proliferation [19]. All this evidence suggests that human molecular phenomena such as inflammation, oxidative stress, and DNA damage can collectively influence dysbiosis and, in turn, have an effect on CRC carcinogenesis. Dysbiosis could add to CRC risk by shifting the effect of dietary elements by promoting a colonic neoplasm together with interacting with gut microbiota. As a consequence, it may be argued that dietary intervention and gut microbiota modulation could have a central role in reducing CRC risk [20].

## 4. Role of Butyrate in Colorectal Cancer Prevention

Many studies have shown how butyrate may prevent later stages of cancer development; this SCFA can be found in cereals like wheat bran and represents the preferred energy substrate for normal colonocytes, being able to induce apoptosis or slow proliferation and differentiation of colon cancer cells [7,21].

Due to these properties, this compound can act as a Histone Deacetylase (HDAC) inhibitor, stopping cell proliferation or inducing apoptosis. In a physiological situation, low doses of butyrate are not sufficient for HDAC inhibition, and consequently, this promotes histone acetylation, leading to colonic cell proliferation. In particular, butyrate is metabolized in the mitochondria, where it is transformed into acetyl-CoA by beta-oxidation. Then, acetyl-CoA condenses with oxalacetate, forming citrate as a result. Citrate goes out of the mitochondria and interacts with the enzyme ATP citrate lyase (ACL), being reconverted into cytosolic and nuclear acetyl-CoA, useful for lipid biosynthesis and as a cofactor for histone acetyltransferases, respectively. On the contrary, high doses of butyrate promote HDAC inhibition, with the final consequence of promoting cell apoptosis. In this case, this compound is not metabolized in mitochondria, but migrates into the nucleus, acting as an HDAC inhibitor; in this way, histones cannot alter the structure of nucleosomes, stopping colonic cell proliferation and increasing apoptosis [21].

In the colorectal cancer scenario, cells undergo the Warburg effect, the mechanism by which tumoral cells prefer glycolysis instead of oxidative phosphorylation, as a source of energy. As a consequence, even lower doses of butyrate do not enter mitochondria, but accumulate in the nucleus of cancerous colonocytes, acting as HDAC inhibitors and apoptosis promoters.

Additionally, another important function of butyrate related to colorectal cancer is its anti-inflammatory action. SCFAs can be ligands for G-coupled receptors called GPR, which regulate different pathways. In particular, butyrate can bind the receptor GPR43 located on the surface of T cells, promoting an anti-inflammatory effect, positive for colorectal cancer prevention [21].

Alvandi et al. designed a study to better understand the link between SCFA concentration in CRC risk and incidence. The analyses have been divided on the available evidence into two outcomes: (1) CRC-risk and (2) incidence. Combined analysis of acetic, propionic, and butyric acid revealed significantly lower concentrations of these SCFAs in individuals with a high risk of CRC (Standardized Mean Difference—SMD = 2.02, 95% CI 0.31 to 3.74, *p* = 0.02). Additionally, CRC incidence was higher in individuals with lower levels of SCFAs (SMD = 0.45, 95% CI 0.19 to 0.72, *p* = 0.0009), compared to healthy individuals. Qualitative analyses identified 70.4% of studies reporting significantly lower concentrations of fecal acetic, propionic, and butyric acid or total SCFAs in those at higher risk of CRC, while 66.7% reported significantly lower concentrations of fecal acetic and butyric acid in CRC patients compared to healthy controls. In conclusion, lower fecal concentrations of the three major SCFAs were associated with higher risk of CRC and incidence of CRC [22].

Shuwen et al., in order to verify the inhibitory effect of sodium butyrate (NAB) and oxaliplatinum (OXA) at the animal level, created a murine model in which subcutaneous implantation of CRC cells was performed and 16S sequencing technology was used to detect intestinal bacteria. GC-MS was used to detect metabolites in mouse stools. NAB was a differential metabolite that affected the efficacy of OXA. It was found that NAB and oxaliplatin could synergically inhibit cell proliferation, migration, and invasion and induce cell apoptosis in cell lines, while animal experiments confirmed the inhibitory effect of oxaliplatin and sodium butyrate on tumors in mice. Furthermore, the intestinal microbe detection and microbial metabolite detection in fecal samples from mice showed a significant increase in species producing butyrate such as Bacteroides, especially when NAB and OXA were administered in combination [23].

In a study performed by Kang et al., an azoxymethane/dextran sodium sulfate-induced mouse CRC model was used to explore the role and mechanism of butyrate in regulating colon cancer and its intestinal microecological balance. Butyrate alleviated weight loss, disease activity index, and survival in CRC mice and inhibited tumor number and progression. Further research revealed that butyrate restrained the aggregation of harmful flora while promoting the colonization of beneficial flora, such as Actinobacteriota, Bifidobacteriales, and Muribaculacea, through 16S rDNA sequence analysis. This study confirmed that butyrate can ameliorate CRC by repairing intestinal microecology, providing ideas and evidence for chemical prophylactic agents, such as butyrate, to counteract tumor growth and regulate tumor microbiota [24].

Garavaglia et al. studied how mutations in APC (Adenomatous Polyposis Coli) are reflected in β-Catenin hyperactivation and loss of proliferation control. Certain intestinal bacteria metabolites have shown the ability to limit CRC cell proliferation and CRC pathogenesis. This study investigated the molecular mechanism underlying the anti-proliferative activity of butyrate, a microbiota-derived short-chain fatty acid, in two CRC cell lines, namely HCT116 and SW620, which bear a mutation in β-Catenin and APC, respectively. Butyrate reduced CRC cell proliferation, as witnessed by the downregulation of proliferation markers. In CRC cells, regardless of the mutational state of APC or β-Catenin genes, butyrate caused the autophagy-mediated degradation of β-Catenin, thus preventing its transcriptional activity [25].

## 5. Interaction between Dietary Fiber and Bile Acids

Dietary fibers also have an important role in colorectal cancer prevention, due to their interactions with bile acids. These compounds are amphiphilic molecules synthesized from cholesterol, in the liver. They are involved in the absorption of lipids in the small bowel and also create micelles to solubilize cholesterol. Bile acids are metabolized in the large bowel; bacterial flora helps to deconjugate and to dehydroxylate primary bile acids, generating secondary bile acids, which have been invoked in colonic carcinogenesis. Experimental evidence in humans and animals showed that bile acids are cytotoxic to colonocytes, stimulating cell proliferation [26].

Anyway, dietary fiber intake can counteract this phenomenon. In fact, fiber is involved in the increase in bowel movement, thus acting on an important etiological factor in colon cancer. Also, fibers bind bile acids, modifying the gut–liver axis. This reduces cholesterol levels, also decreasing the risk of colorectal cancer growth [15,27].

From a molecular point of view, fibers entrap polyphenols, promoting their bioavailability for gut bacteria metabolism. Indeed, the gut microbiota can modulate oxidative stress positively or negatively, due to the degradation of fibers in phenolic compounds. These compounds act as antioxidants after being absorbed by the intestine and sent to the bloodstream [15]. Figure 1 illustrates interactions between dietary fiber and gut microbiota.

## 6. Water-Holding Capacity of Fibers

The last beneficial feature of dietary fibers concerns specifically soluble ones. It is shown, in fact, that these molecules, at low concentrations, have great water-holding capacity by forming polysaccharidic gel. In opposition to fruit and vegetable cells, wheat bran cells have lignified walls; this confers to wheat bran gel walls a lower water-holding capacity. The hydrophobic characteristics of these cell walls are important for cancer prevention. Indeed, carcinogens can easily bind to hydrophobic walls, so that they can pass out of the alimentary tract in the stools, reducing their interaction with colonic mucosal cells. Because of their indigestibility, dietary fibers can also increase the bulk of the feces, with the consequence of their shorter transit time in the gut. In this way, there will be a lower concentration of carcinogens interacting with colonic mucosal cells [7].

The main mechanisms by which fibers may counteract CRC development are summarized in Figure 2.

## 7. Fiber in the Prevention of CRC

World Cancer Research Fund/American Institute for Cancer Research (WCRF/AICR) suggest that decreased consumption of grains and dairy products and the Western diet, characterized by a high consumption of red, processed meat and fats, increase the risk of CRC [29].

The hypothesis of the protective role of fiber against CRC started in 1971 from Burkitt, who reported that colorectal cancer was rare among rural Africans. In fact, the incidence of colorectal cancer in men 35–64 years of age was 3.5/100,000 in Kampala (Uganda), as opposed to 51.8/100,000 in Connecticut (USA). These data were explained by the fact that Africans had small amounts of meat and a lot of fiber from fruits, grains, and vegetables in their diet [30].

In recent decades, many clinical trials have investigated the topic and showed that the prevalence of CRC increases inversely to the intake of dietary fiber, with few exceptions. Table 2 shows the decrease in incidence of adenomas associated with a high-fiber diet given to patients, in many clinical trials. Anyway, some US studies have reported no association. Conversely, the European Prospective Investigation into Cancer and Nutrition (EPIC) study (1992–2015), designed to investigate the relationships of diet and environmental factors with the incidence of cancer, demonstrated a 40% reduction in CRC risk in the highest quintile of fiber intake compared with the lowest [31]. In a study by Yu et al. [32], non-starch polysaccharides, which are a type of soluble fiber, were inversely correlated with CRC onset after more than twenty years. In particular, a dose-dependent relationship was observed. Patients consuming the highest quartile of fibers showed a hazard ratio of 0.84, compared with the lowest quartile. Such an inverse association was more evident for colon cancer rather than rectal cancer. On the other hand, in the Polyp Prevention Trial, which randomized the consumption of a high-fiber, high-fruit and vegetable, and low-fat diet for 4 years to evaluate colorectal adenoma recurrence, it was found that, despite the fact that this intervention did not influence serum bile acid levels, high baseline values of bile acid were associated with increased risk of adenoma recurrence, with an OR = 2.17 [33]. The PrebiotiCa study [34] investigated six different soluble, prebiotic fibers (including nystose, kestose, 1F-β-fructofuranosylnystose, raffinose, and stachyose) and found the strongest preventive effect for raffinose, with an OR of 0.73, and for stachyose, with an OR of 0.64, by comparing the highest to the lowest quintile. The preventive link with stachyose was more substantial for colon (OR = 0.74) than for rectal cancer. In a ten-year study conducted on patients with Lynch syndrome, who have a gene-related higher risk of CRC, patients were randomized to consume 30 g resistant starch daily or placebo for up to 4 years. No effect was observed for the incidence of CRC, while a protective effect towards extra-colonic, Lynch-syndrome-related cancers was reported [35]. The heterogeneity between studies in the US and in Europe might be explained considering the different food sources of fiber (cereals vs. fruits and vegetables) and the overall lower fiber intake in the US cohorts. Furthermore, several studies have associated whole grains and reduced risk of CRC [36]. In particular, a 2011 meta-analysis found a 10% reduction in CRC risk for a 10 g/day intake of dietary fiber, and approximately a 20% reduction for a 90 g/day intake of whole grains [37].

A 2018 meta-analysis conducted by Yu Ma et al. investigated the relationship between dietary fiber intake and subsite-specific colorectal cancer. In fact, CRC can be classified into two different types, in relation to the site: proximal colon cancer (PCC) and distal colon cancer (DCC), with different genetic and environmental risk factors. The result of the analysis was that fiber intake was inversely associated with the risk of CRC in both subgroups, with no significant heterogeneity. The risk ratio (RR) of PCC for individuals with the highest fiber intake compared to those with the lowest intake was 0.86; for DCC the RR was 0.79, i.e., 21% lower than individuals with the lowest fiber intake [38].

A 2017 update of the Cochrane review has analyzed the effect of dietary fiber on the recurrence of adenomatous polyps and on the incidence of CRC in people with a known history of adenomatous polyps (but no previous CRC) which had been removed to achieve a polyp-free colon at baseline. No statistically significant difference was found between the intervention and control groups for the number of participants with at least one adenoma, more than one adenoma, or at least one adenoma 1 cm or greater at three to four years. After 8 years of comprehensive dietary intervention, no statistically significant difference was found in the number of participants with at least one recurrent adenoma or with more than one adenoma. However, authors themselves claimed that the results should be interpreted cautiously because of the high rate of loss to follow-up and also because adenomatous polyp is a surrogate outcome for the true endpoint, which is CRC [39].

The Polyp Prevention Trial (PPT), a multicenter randomized clinical trial, evaluated the effects of a high-fiber, high-fruit and vegetable, and low-fat diet on the recurrence of adenomatous polyps in the large bowel over a period of 4 years. Similarly, the risk of recurrent adenomas was not significantly different from that of the controls. The PPT-Continued Follow-up Study (PPT-CFS) was a post-intervention observation of PPT participants for an additional 4 years from the end of the trial; the results showed no significant intervention–control group differences in the relative risk for recurrence of an advanced adenoma or multiple adenomas [40].

On the bases of these results, we can hypothesize that dietary fiber is able to prevent the evolution of polyps into CRC more than their onset.
ijms-24-13501-t002_Table 2Table 2Clinical trials showing the association between CRC and high-fiber diet.Author, YearType of TrialResultsFibresAlberts, 1996 [41]Randomized, double-blinded trialAt 9 months, high-dose fiber supplementation caused 52% reduction (*p* = 0.001) in fecal concentrations of total bile acidsdietary wheat bran fiber (2.0 or 13.5 g/day) in the form of cereal and supplemental calcium carbonate (250 or 1500 mg/day elemental calcium)Alberts, 2000 [42]Randomized trialOR (high-fiber group) = 0.88 (*p* = 0.28)OR (low-fiber group) = 0.99 (*p* = 0.93)High amounts/low amounts of wheat bran fiberBonithon-Kopp, 2001 [43]Randomized blinded, placebo-controlled trial45% increase in recurrent adenomas relative to the placebo group at 3 years (*p* = 0.04)/MacLennan, 1995 [44]Randomized, partially double-blinded, placebo-controlled factorial trialAt 24 months, OR = 0.4At 48 months, OR = 0.325 g of wheat bran daily and a capsule of beta carotene (20 mg daily)Schatzkin, 2000 [45]Randomized trialIn the control group, 39.7% and 39.5% had one recurrent adenoma (OR = 1)High-fiber diet (18 g of dietary fiber per 1000 kcal)


## 8. Conclusions

Since CRC is one of the types of cancer most affected by diet, a healthy diet can be essential to set up its natural and permanent chemoprevention. Fiber is an essential component of a healthy diet and its beneficial effects in preventing the development of CRC have been clearly elucidated in several studies. These benefits can be obtained through various mechanisms, which contribute to their achievement. On these bases, it appears evident that a correct nutritional education could be fundamental for reducing the incidence of CRC in populations at risk. This education should start from childhood in order to promote the prevention not only of metabolic diseases, but also of neoplastic diseases more influenced by dietary factors [46]. In this regard, dietary fiber supplementation could be a strategy to propose, although its clear role in chemoprevention has been widely encouraged, but not clearly demonstrated by studies on large population samples.

Finally, it should be mentioned that the waste of fibers is a topic that deserves further discussion, since it may underpower the beneficial effect on CRC. Indeed, it is known that different types and sources of dietary fibers can derive from agri-food by-products. Food processing industrial wastes include bio-actives such as dietary fibers, pigments, essential minerals, fatty acids, and antioxidant polyphenolic compounds. Therefore, ample amounts of fibers that derive from non-edible and edible parts of fruits and vegetables are often wasted along the entire agri-food supply chain [47,48,49].

## Figures and Tables

**Figure 1 ijms-24-13501-f001:**
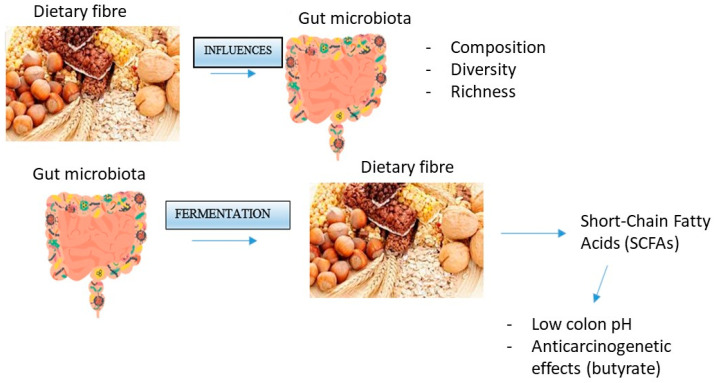
Interactions between dietary fiber and gut microbiota [28].

**Figure 2 ijms-24-13501-f002:**
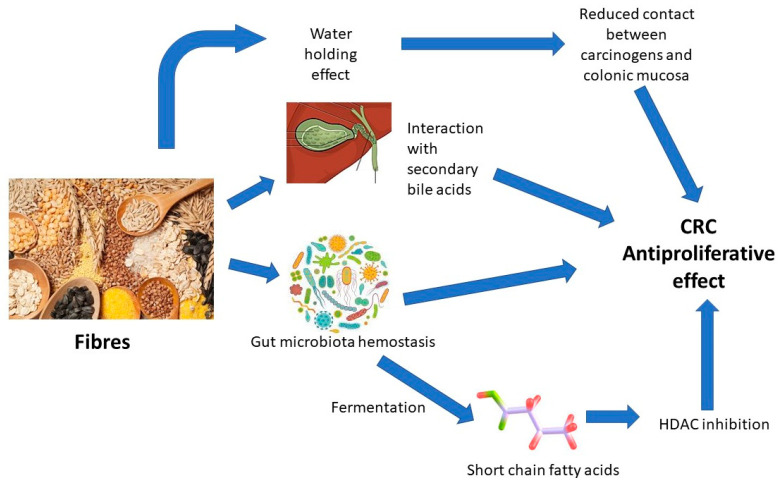
Effect of dietary fiber in the regulation of epithelial homeostasis and anti-carcinogenetic properties.

## Data Availability

Not applicable.

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
