# Peer review of "Fibres and Colorectal Cancer: Clinical and Molecular Evidence"

_ijms, 2023, doi:10.3390/ijms241713501_

Round 1

Reviewer 1 Report (Previous Reviewer 3)

The article "Fibres and Colorectal Cancer: Clinical and Molecular Evidence" endeavors to provide evidence in favor of the consumption of dietary fiber for the purpose of combating colorectal cancer. However, upon examination, it becomes clear that the author has made only superficial changes and added a few additional authors, which is the only positive aspect of the article. Regrettably, the authors seem to have disregarded prior peer reviews and neglected to address major concerns that were raised in the previous review. Additionally, the manuscript lacks novelty and fails to cover other aspects related to the proposed topic. The included picture and tables appear amateurish, and the authors exhibit unprofessionalism in their drafting and editing, evidenced by uneven fonts. Taken together, the present manuscript is a disappointment and is unlikely to appeal to the readership of the International Journal of Molecular Science. No significant attempt has been made to improve upon the prior submission with Manuscript-ID: ijms-2296226

No comments at this stage of the manuscript. 

Author Response

We are sorry about the comment of the reviewer. However, we do not agree that we have disregarded the indications of the referees. Indeed, as indicated in the previous rebuttal letter, we have added an additional figure and an additional table, and added a paragraph about fibre wasting, as requested.

We do not have a professional service for figure editing and this is why figures may seem amateurish, but we believe that they still are able to give the correct message.

Reviewer 2 Report (Previous Reviewer 1)

The manuscript is improved in this form.

Please, correct only some sentences, typos, and English language in some parts of the manuscript. 

Minor corrections are needed (typos, spelling)

Author Response

We thank the reviewer for the kind comment. We checked the paper and corrected some typos and grammar errors.

This manuscript is a resubmission of an earlier submission. The following is a list of the peer review reports and author responses from that submission.

Round 1

Reviewer 1 Report

The authors focus their review titled “Fibres and colorectal cancer: clinical and molecular evidences on the benefit and strength of the use of fibres in diet, with particular attention to colorectal cancer.

The manuscript is easy to follow. However, I have some concerns:

-        The references are not always present. In some cases, they lack.

From lines 99-105 please add more references.

In the 1st paragraph, authors could add a table resuming the different classifications of the fibres (soluble vs insoluble).

-        Figure 1: please improve it.

-        Please provide a figure resuming “Interaction between fibre and the microbiota”

The authors describe the importance of fibres in the prevention of CRC. Are there present clinical trials based on this finding? Please discuss or add a table relative to clinical trials.   

It is known that different types and sources of dietary fibres can derive from agri-food by-products.  Food processing industrial wastes include bio-actives such as dietary fibres, pigments, essential minerals, fatty acids, antioxidant polyphenolic compounds, etc., and require green approaches to obtain these value-added compounds. In their conclusions, authors could highlight the ample amounts of fibres that derive from non-edible and edible parts of fruits and vegetables and that are wasted along the entire agri-food supply chain. This is a very current and appealing topic. Please consider: Sustainability 202012(13), 5401; https://doi.org/10.3390/su12135401; Molecules 202025(3), 510; https://doi.org/10.3390/molecules25030510; Cancers (Basel). 2022 Nov 10;14(22):5517. doi: 10.3390/cancers14225517

Reviewer 2 Report

The information is of interest, however, there are many publications with information on dietary fiber and colorectal cancer and its molecular mechanisms.

What does this publication offer as opposed to the existing ones? 

A similarity check was performed on the Turnitin software finding complete paragraphs the same as other publications, for example: Lines from 99 to 119, 138 to 150, 198 to 231.

Turnitin report attached

Reviewer 3 Report

The review entitled "Fibres and colorectal cancer: clinical and molecular evidences" attempts to compile evidence that suggests dietary fiber intake to fight colorectal cancer. However, the authors fail to make an impression nor covered other aspects related to the proposed topic. The manuscript title starts with a spelling error Fibre and is filled with various ambiguous statements "page 2 line83 Dietary fibers are an environmental factor associated with the risk for colorectal cancer." The manuscript did not make any attempt to add any new information to the existing body of information. The picture made for this article is not professionally done and there are no tables to describe a previously published work regarding the topic. Please follow some good examples of how to present your topic  E.g., https://www.mdpi.com/1422-0067/24/4/3744. The present manuscript lacks novelty and is not in a form that could be appealing to the International Journal of Molecular science reader base. If needed to be considered the authors need to be more dedicated to redrafting the manuscript, adding interesting perspectives related to the topic, rephrasing to make the manuscript to appeal a global audience, revising the language, and resubmitting for peer reviewing. Good luck